# Mucosal Vaccination with UV-Inactivated *Chlamydia suis* in Pre-Exposed Outbred Pigs Decreases Pathogen Load and Induces CD4 T-Cell Maturation into IFN-γ^+^ Effector Memory Cells

**DOI:** 10.3390/vaccines8030353

**Published:** 2020-07-02

**Authors:** Amanda F. Amaral, Khondaker S. Rahman, Andrew R. Kick, Lizette M. Cortes, James Robertson, Bernhard Kaltenboeck, Volker Gerdts, Catherine M. O’Connell, Taylor B. Poston, Xiaojing Zheng, Chuwen Liu, Sam Y. Omesi, Toni Darville, Tobias Käser

**Affiliations:** 1Department of Population Health and Pathobiology, College of Veterinary Medicine, North Carolina State University, 1060 William Moore Drive, Raleigh, NC 27607, USA; afamaral@ncsu.edu (A.F.A.); arkick@ncsu.edu (A.R.K.); lmlorenz@ncsu.edu (L.M.C.); 2Comparative Medicine Institute, North Carolina State University, 1060 William Moore Drive, Raleigh, NC 27607, USA; 3Department of Pathobiology, College of Veterinary Medicine, Auburn University, Auburn, AL 36849, USA; ksr0003@auburn.edu (K.S.R.); kaltebe@auburn.edu (B.K.); 4College of Veterinary Medicine, North Carolina State University, 1060 William Moore Drive, Raleigh, NC 27607, USA; cvmstatistician@ncsu.edu; 5Vaccine and Infectious Disease Organization—International Vaccine Centre (VIDO-InterVac), University of Saskatchewan, 120 Veterinary Road, Saskatoon, SK S7N 5E3, Canada; volker.gerdts@usask.ca; 6Department of Pediatrics, University of North Carolina at Chapel Hill, Chapel Hill, NC 27599, USA; catherine.oconnell@unc.edu (C.M.O.); tbposton@email.unc.edu (T.B.P.); xiaojinz@email.unc.edu (X.Z.); somesi@email.unc.edu (S.Y.O.); lad@email.unc.edu (T.D.); 7Department of Biostatistics, University of North Carolina Gillings School of Global Public Health, Chapel Hill, NC 27599, USA; chengwen_li@med.unc.edu

**Keywords:** *chlamydia trachomatis*, *chlamydia suis*, large animal model, swine, translational research, vaccine development, TriAdj, vaccination, immunology, T cells, effector memory

## Abstract

*Chlamydia trachomatis* (*Ct*) infections are the most frequent bacterial sexually transmitted disease, and they can lead to ectopic pregnancy and infertility. Despite these detrimental long-term sequelae, a vaccine is not available. Success in preclinical animal studies is essential for vaccines to move to human clinical trials. Pigs are the natural host to *Chlamydia suis* (*Cs*)—a chlamydia species closely related to *Ct*, and are susceptible to *Ct,* making them a valuable animal model for *Ct* vaccine development. Before making it onto market, *Ct* vaccine candidates must show efficacy in a high-risk human population. The high prevalence of human *Ct* infection combined with the fact that natural infection does not result in sterilizing immunity, results in people at risk likely having been pre-exposed, and thus having some level of underlying non-protective immunity. Like human *Ct*, *Cs* is highly prevalent in outbred pigs. Therefore, the goal of this study was to model a trial in pre-exposed humans, and to determine the immunogenicity and efficacy of intranasal *Cs* vaccination in pre-exposed outbred pigs. The vaccine candidates consisted of UV-inactivated *Cs* particles in the presence or absence of an adjuvant (TriAdj). In this study, both groups of vaccinated pigs had a lower *Cs* burden compared to the non-vaccinated group; especially the TriAdj group induced the differentiation of CD4^+^ cells into tissue-trafficking CCR7^-^ IFN-γ-producing effector memory T cells. These results indicate that *Cs* vaccination of pre-exposed pigs effectively boosts a non-protective immune response induced by natural infection; moreover, they suggest that a similar approach could be applied to human vaccine trials.

## 1. Introduction

*Chlamydia trachomatis* (*Ct*) continues to be the most prevalent sexually transmitted disease worldwide [1,2]. Especially after repeated or long-term genital infections, *Ct* can lead to pelvic inflammatory disease [3]. Thereby, *Ct* contributes to relevant reproductive issues such as ectopic pregnancy and infertility. Despite these detrimental long-term sequelae and extensive research into *Ct* vaccine development, a *Ct* vaccine is not available. 

The lack of a *Ct* vaccine can be partially explained by limitations of the currently used animal models (Figure 1). The advantages and disadvantages of different animal models for *Ct* research has been recently reviewed [4]. The mouse model has advantages of low cost and a vast toolkit; however, important differences between mice and humans limits its use as a translational animal model. A high number of 185 immune genes are not shared with humans [5] and IFN-γ’s inhibitory effects on *Ct* differ in humans and mice. Since IFN-γ is a key immune modulator in the response against *Ct*, this difference impairs the use of mice for *Ct* research [6]. Nevertheless, the mouse model was used for possibly one of the most promising *Ct* vaccination studies. Stary et al. showed in 2015 that mucosal administration of UV-inactivated *Ct* combined with charge-switching synthetic adjuvant particles (cSAP) resulted in long-lived protection against genital *Ct* challenge. In addition, they demonstrated that the induction of two subsets of IFN-γ-producing CD4^+^ T cells is crucial for protection against *Ct*—tissue-resident memory T cells (T_RM_) and circulating memory T cells consisting of both lymph node-trafficking central memory (T_CM_) and tissue-trafficking effector memory (T_EM_) T cells [7].

Non-human primates (NHP) are very similar to humans, but NHP studies are expensive and involve ethical concerns. As a result, NHPs were only used in six *Ct* vaccine development studies, which makes NHP only the fourth most frequently used animal model [8]. The limitations of these two animal models have led to a bottleneck for *Ct* vaccine development (Figure 1). Consequently, *Ct* vaccine researchers are exploring the use of alternative animal models such as guinea pigs, koalas, and pigs. 

With eight chlamydia vaccine studies, koalas are the second most frequently used animal model. However, these studies are concentrated on *C. pecorum*, which naturally infect koalas and lead to devastating reproductive issues. Thus, the end goal for many koala vaccine studies is protection of this iconic Australian marsupial. 

The third most frequently used animal model for *Ct* vaccine development is the pig—both commercial and minipig breeds. The pig has several advantages as a biomedical translational animal model [4,9] and it significantly contributed to the study of sexually transmitted diseases including *Ct* vaccine development [10,11,12,13,14,15]. Pigs have a similar size, physiology, and hormonal reproductive cycle to humans [16]. Pigs are not only susceptible to common *Ct* strains, but they are also the natural host for *Chlamydia suis* [17]—a chlamydia species very close to *Ct* [18,19]. Finally, pigs have an immune system that is very similar to humans. Compared to mice, pigs have more immunological genes shared with humans (230 genes), and 4.5 times fewer genes (41 genes) that are not shared with humans [5].

Although ethical concerns exist for any animal trial, the pig has the advantage of being used as a food animal. Porcine genital tracts, blood, and lymph nodes are by-products of meat production. Thereby, primary cells used for biomedical *Ct* research can be collected at nearly no cost and without sacrificing animals—a strong advantage in accordance with the 3R principle—replacement, refinement, and reduction. Furthermore, *Ct* vaccine trials in pigs cost substantially less than comparable trials in NHPs [4].

Pigs have been used to study *Ct* infections and vaccine development since 2005 [15]. The Vanrompay group showed that *Ct* can successfully infect specific-pathogen-free (SPF) pigs [15], and then tested vaccine candidates in SPF pigs [12,13,14]. They were also able to isolate *Cs* out of humans—a sign for the zoonotic potential of the pig pathogen *Cs* [18]. A strong Danish collaboration of the Agerholm, Anderson, Follmann, and Jungersen research groups used Goettingen minipigs for their *Ct* research. This collaboration studied not only basic characteristics of the pig model relevant for translational *Ct* research, they also used their combined expertise for *Ct* vaccine development and demonstrated *Ct* vaccine efficacy and immunogenicity in naïve minipigs [10]. In their latest studies, the authors showed that protection against *Ct* genital infection in minipigs immunized with *Ct* vaccine formulated with CAF01 adjuvant was associated with cervical infiltration of CD4^+^ T cells [20] and tissue-resident memory CD4^+^ T cell infiltration into the uterus [21]. Käser et al. provided a detailed view of the porcine T-cell immune response to *Cs* and *Ct*. As in humans, pigs develop a strong CD4^+^ T-cell response upon *Ct* or *Cs* infection consisting of mainly IFN-γ single- or IFN-γ/TNF-α- double-cytokine producing CD4^+^ T cells; this response was heterologous, thus, *Cs* infected pigs responded to *Ct* and vice versa [11].

This heterologous response provides a great advantage for the pig model, since *Cs* is highly prevalent in outbred pigs. Vaccination of *Cs*-pre-exposed outbred pigs can simulate *Ct* vaccination of pre-exposed humans. This gives this animal model the ability to predict the outcome of phase III clinical vaccine trials in pre-exposed humans. This last pre-licensing phase is the most time-consuming, extensive, and costly vaccine development phase. To facilitate timely completion for STIs, phase III trials will require completion in high-risk populations. Thus, any successful *Ct* vaccine candidate must show safety, immunogenicity, and efficacy in pre-exposed humans of high genetic diversity. A vaccine candidate that shows immunogenicity and efficacy in this *Cs* pre-exposed animal model may also be immunogenic and efficacious in *Ct* pre-exposed humans. Furthermore, once a *Ct* vaccine successfully completes the clinical trial phases and makes it to the market, it should be administered to as many individuals as possible. Wide coverage will optimize herd immunity against *Ct*. Due to the high prevalence of *Ct*, it would be detrimental to vaccinate only naïve patients. However, a model for testing *Ct* vaccines in pre-exposed outbred animals to closely resemble the situation in humans is currently not available. To overcome that limitation and to provide essential information on the effect of *Ct* vaccines in pre-exposed patients, the goal of this study was to establish a model for testing vaccine immunogenicity and efficacy in outbred, pre-exposed pigs. 

Therefore, we performed a proof-of-principle *Cs* vaccine study in outbred commercial high-health pigs with documented pre-exposure to *Cs*. Before vaccination, these pigs received antibiotic treatment to eliminate genital *Cs* infection. Pigs received two intranasal vaccinations of either MOCK or UV-inactivated *Cs* particles with or without the TriAdj adjuvant [2]. Sixteen days post-vaccination, pigs were challenged post-cervically with *Cs*. Throughout the study, *Cs* load was evaluated via qPCR, and the induced immune response was monitored by ELISA and multi-color flow cytometry. We determined that prime/boost intranasal administration of a killed whole-cell *Cs* vaccine is both immunogenic and effective. This vaccine strategy induced the differentiation of IFN-γ-producing CD4^+^ cells into tissue-trafficking T-effector memory cells; moreover, compared to MOCK-vaccinated pigs, it effectively limited genital *Cs* infection. This study demonstrates that outbred pre-exposed pigs can serve as a valuable animal model for *Ct* vaccine development.

## 2. Materials and Methods 

### 2.1. Chlamydia Suis

The *Chamydia suis* strain S45 (ATCC VR-1474 strain 545 lot 1171210) was propagated in HeLa cells using standard technique [22] and purified as previously described [23]. Bacteria were titrated on HeLa cells as previously described [24].

### 2.2. Vaccine and Adjuvant Preparation 

The vaccine used in this study consists of ultraviolet light (UV) inactivated *Chlamydia suis* only or in association with the triple adjuvant combination (TriAdj [1], Vaccine and Infectious Disease Organization—International Vaccine Center (VIDO-InterVac), Saskatchewan, Canada). Each vaccine dose includes 1 × 10^9^
*Cs* inclusion-forming units (IFU) in 1 mL of sucrose phosphate glutamic acid buffer (SPG, [22]) and exposed to 8 watts UV light at 30 cm distance for 1 h (adapted from [7]). The TriAdj was prepared according to the manufacturer’s instructions. The final composition per pig was 150 µg of poly I:C; 300 µg of host defense peptide; and 150 µg of polyphosphazene. 

### 2.3. Pigs and Experimental Design 

Twenty-four 25-week-old sexually mature female *Cs* pre-exposed pigs were selected from a commercial high-health farm. Their *Cs* exposure status was determined by qPCR of vaginal swabs as described below. The setup and timeline of this study is shown in Figure 2. Upon arrival, pigs were randomly distributed into four groups with six pigs each. To treat the *Cs* infection, each pig was treated daily by oral administration of 1.44 g of doxycycline (Doxycycline Hyclate, West-Ward, Eatontown, NJ, USA) for four days and additionally with 3 g of tylosin (Tylan soluble, Elanco^TM^, Indianapolis, IN, USA) twice a day for 3.5 days. 

Pigs were vaccinated intranasally with 1.5 mL/nostril of either a control solution (SPG; groups “MOCK” and “*Cs-chall*”) or with 10^9^ UV-inactivated *Cs* particles without adjuvant (group “*Cs*-chall + vacc”) or with adjuvant (group “*Cs*-chall + TriAdj vacc”) in SPG at 0 and 14 days post first vaccination (dpv). Intranasal vaccination was performed using an intranasal mucosal atomization device (MAD Nasal^TM^ Mist, Teleflex medical, Research Triangle Park, NC, USA).

From 10–23 dpv, the estrus cycle of pigs was synchronized with synthetic progesterone according to manufacturer instructions (MATRIX®, Merck, Madison, NJ, USA). At 30 dpv, pigs were challenged trans-cervically with SPG (group MOCK) or SPG with 10^8^
*Cs* particles (all “*Cs*-chall” groups)—a standard *Cs* challenge dose for (mini-) pigs [12,13,15,25]. Transcervical challenge was performed in a total volume of 20 mL using gilt post cervical artificial insemination (PCAI) catheters (kindly provided by IMPORT-VET, Centelles, Spain) connected to a 50 mL syringe.

Animals were clinically monitored throughout the study including hyperthermia and vaginal discharge. Pigs were considered to have hyperthermia if their rectal temperature was ≥39.5 °C [26]. Blood and swabs were collected as shown in Figure 2. Pigs were sacrificed using captive bolt gun followed by exsanguination. These procedures are in accordance with and approved by the North Carolina State University Institutional Animal Care and Use Committee (IACUC #17-029-B). 

### 2.4. Sampling 

Swab samples were collected using 4NG FLOQ swabs (Copan flock technologies, Murrieta, CA, USA) as described previously [12]. Blood samples for serum and peripheral blood mononuclear cell (PBMC) isolation were collected from the external jugular vein into SST or heparin tubes, respectively. Serum was incubated at room temperature for 2 h and spun at 2000× *g* for 20 min at 23 °C. Isolation of PBMC was performed using lymphocyte separation medium (Ficoll-Paque Premim, density 1.077 g/mL, GE Healthcare, Uppsala, Sweden).

### 2.5. Detection of Chlamydia via qPCR

Chlamydia DNA from vaginal swabs was used to measure the chlamydia infection load via Taqman qPCR assay with primers and probe targeting the 23S rRNA gene of *Cs* as previously described including primer and probe design [23]: Fwd primer: CCTAAGTTGAGGCGTAACTG, Rv primer: GCCTACTAACCGTTCTCATC, Probe: FAM-TTAAGCACGCGGACGATTGGAAGA-TAMRA. The Taqman qPCR was run on a qTOWER3G qPCR machine (AnalytikJena, Jena, Germany). A standard curve of *Cs* gBlocks (IDT® Integrated DNA Technologies, Coralville, IA, USA) was included on every plate to determine the number of chlamydia particles per swab.

### 2.6. Serum Anti-Chlamydia Suis Immunoglobulin G Detection

Primary IgG antibodies were detected with horseradish peroxidase (HRP)-conjugated goat anti-pig IgG-h+l cross-adsorbed antibody (A100-205P) (Bethyl Laboratories, Inc., Montgomery, TX, USA) by colorimetric enzyme-linked immunosorbent assay (ELISA) as described previously [27], with two modifications: (A) wash buffer contained 200 mM NaCl, 0.1% Tween-20, 0.005% Benzalkonium chloride, 20 mM Tris-HCl, and a pH of 7.4; (B) assay diluent and blocking buffer contained 10% chicken sera, 1% polyethylene glycol, 200 mM NaCl, 0.1% Tween-20, 0.005% Benzalkonium chloride, 50 mM Tris-HCl, and a pH of 7.4. 

Streptavidin-coated microtiter plates (Thermo-Fisher Scientific, Waltham, MA, USA, Nunc #436014) were coated with two mixtures of *Cs*-specific biotinylated peptide antigens: (1) 18 peptides from CT442, CT529, CT618, OmcB, OmpA, and PmpD proteins; and (2) 18 peptides from IncG, IncA, and two IncA family proteins. Non-coated wells served as background controls. Background-corrected colorimetric signals (OD values) of each individual serum was calculated by subtracting 120% serum background (mean + 2 × SD) produced in non-coated wells from the signals produced in wells coated with the two *Cs*-specific peptide mixtures. Average OD of two background-corrected signals for these *Cs*-specific peptide mixtures was used in the analysis to determine IgG level in the serum. 

### 2.7. Neutralizing Antibody Detection 

Neutralizing antibody detection was performed as previously described [11]. Shortly, serum was heat-inactivated at 56 °C for 45 min and incubated with *Cs* (MOI of 0.5) at 37 °C for 30 min in a 1:10 final serum dilution. Next, confluent HeLa cells were infected with this serum-*Cs* mix by centrifugation at 900× *g* for 1 h at 37 °C. After an additional hour of incubation, cells were washed and incubated for 30 h. Then, cells were harvested and stained for flow cytometry evaluation of infection using the anti-chlamydia antibody clone ACI (LSBio, Seattle, WA, USA) and the secondary anti-mouse IgG3-Alexa 488 antibody (Southern Biotech, Birmingham, AL, USA). Cells were recorded on a Cytoflex flow cytometer using the CytExpert software (Beckman Coulter, Brea, CA, USA). Data analysis was performed with FlowJo version 10.5.3 (FLOWJO LLC, Ashland, OR, USA) as previously described [24]. Percent suppression was calculated for each animal using the following formula: $$\%\;\mathrm{suppression}=100-\left(\frac{\%\;\mathrm{infection}\;\left[x\;\mathrm{dpc}\right]}{\%\;\mathrm{infection}\;\left[-2\;\mathrm{dpc}\right]}\right)\times 100$$
where “x dpc” (days post challenge) is the day of the calculated percent of suppression.

### 2.8. Chlamydia suis-Specific CD4 T-Cell Proliferation

Thawed PBMCs were stained with CellTrace^TM^ Violet and seeded in microtiter plates (Sarstedt, Nümbrecht, Germany) in quadruplicates at a density of 2 × 10^5^ cells/well in RPMI-1640 (Corning, Corning, NY, USA) supplemented with 10% FBS (VWR, Radnor, PA, USA) and 1× antibiotic-antimycotic (Corning). Cells were co-cultured for 4 days with *Cs* lysate at 1 µg/ml. After cultivation, quadruplicates were pooled and stained according to Table 1. Cells were recorded and data was analyzed as described in the previous section. The gating hierarchy used for this analysis is shown in Appendix A. In addition to a standard percentage analysis of cell subsets, we are introducing two novel ways of numerical and statistical evaluation of the immune response to a pathogen/vaccination: (I) The “Differentiation value”: To facilitate the statistical comparison of the differentiation status of each group, we first gave T_naïve_ cells a value of zero, T_CM_ a one, and T_EM_ a two, based on their differentiation status: T_naïve_ are undifferentiated (T_naïve_ = 0), then they first differentiate into T_CM_ (T_CM_ = 1), and then into T_EM_ cells (T_EM_ = 2). Then, we multiplied the frequency of each of the three differentiation statuses with the respective value: for example, if 20% of the proliferating CD4 T cells in an animal are naïve, 30% are T_CM_, and 50% are T_EM_, the differentiation value for these proliferating CD4 T cells is calculated as follows—(T_naïve_) 0.2 × 0 + (T_CM_) 0.3 × 1 + (T_EM_) 0.5 × 2 = 1.3. And (II) the “Response value”: Here, we introduce a method for combined analysis of the cellular response (e.g., proliferation or IFN-γ production) with the cell differentiation status (T_naïve_, T_CM_, and T_EM_). To provide this combinatorial response value, we multiply the value of the cellular response (e.g., 15% proliferation) with the previously described differentiation value of this cell subset (e.g., 0.6) and 100. Thus, for this example, the response value is 0.15 × 0.6 × 100 = 9.

### 2.9. Chlamydia suis-Specific IFN-Γ Production by CD4 T Cells

Frozen PBMCs were thawed and stimulated as above but at a density of 5 × 10^5^ cells/well and only for 18 h. Monensin (5μg/mL, Alfa Aesar, Ward Hill, MA, USA) was added for the last 4 h of cultivation to block the cellular Golgi export system. Cultured cells were pooled and stained for flow cytometry as mentioned above and as stated in Table 1. Recording and analysis of the flow cytometry data was performed as described above. The gating hierarchy used for this analysis is shown in Appendix A. “Differentiation value” and “response value” at 37 dpv were calculated for each animal as described in the previous section.

### 2.10. Blood CD4 T-Cell mRNA Data Acquisition, Processing, and Analysis 

Frozen PBMC were thawed and stimulated with *Cs* lysate overnight as described above; only the addition of the Golgi inhibitor was omitted. At the end of the culture, PBMC were harvested and CD4 T cells were isolated by magnetic activated cell sorting (MACS) using positive cell sorting on CD4^+^ cells (Miltenyi Biotech, Bergisch Gladbach, Germany), according to manufacturer’s protocol. The purity of the MACS sorts was confirmed by flow cytometry using antibodies according to Table 1. The sort purity was consistently over 90% in the CD4^+^ cell fraction (Appendix A).

Transcriptional responses of purified CD4 T cells were then profiled using RNA-seq. Cells were stored at −80 °C in preservative solution (RNA/DNA Shield, Zymo Research, Irvine, CA, USA) prior to extraction. Total RNA was extracted using a Quick RNA™ nucleic acid isolation kit (Zymo Research) with on-column DNAse I treatment of the RNA. Library preparation and sequencing was conducted by the High-Throughput Sequencing Facility at the University of North Carolina at Chapel Hill. cDNA libraries were generated from rRNA-depleted template using the NuGen Ovation SoLo RNA-Seq System (NuGen, San Carlos, CA, USA). Following library cDNA quantification, the pooled libraries were sequenced using a SI flow cell on the Illumina Novaseq sequencing platform (50 bp, paired ends). Base calling and quality filtering were performed per the manufacturer’s instructions. 

Quality control and trimming were performed using Fastqc [28] and fastq-mcf [29], respectively. The QC report was first applied to the raw sequence data followed by application of the sequence Trimmer. As a result, sequences were trimmed based on a phred quality score of more than 20 and cycle removal at an ‘N’ (bad read) of 0.5%. The filtered sequences were mapped onto the white pig gene ensemble (Ensembl build 11.1), using Spliced Transcripts Alignment to a Reference (STAR) [30]. For comparing the levels of gene expression across all samples, the read counts per gene were normalized using DESeq2 [31], which rescaled the counts using the relative effective library sizes. Genes with normalized counts greater than or equal to 25 in at least five samples were used for analysis. The differential gene expression between the non-challenged MOCK group and challenged groups (*Cs* challenged, *Cs* challenged + vaccinated, and *Cs* challenged + Tri Adjuvant vaccinated) was assessed using DESeq2. DESeq2 estimated variance-mean dependence in count data from high-throughput sequencing and tested for differential expression based on a model using the negative binomial distribution. Multiple testing was adjusted by Benjamini-Hochberg.

### 2.11. Statistical Analysis

Statistical analyses were performed using GraphPad Prism (GraphPad Software, Inc., La Jolla, CA, USA). Data comparisons at specific time points were analyzed using one-way ANOVA with in vivo infection as the one factor. Data comparisons throughout the study were performed using repeated measures two-way ANOVA with in vivo infection and time as the two factors; post hoc multiple comparisons were performed using the Dunnett’s test. Differences were defined significant (*) for *p* < 0.05.

## 3. Results

The goal of this study was to determine the efficacy and immunogenicity of *Cs* vaccination in pre-exposed outbred pigs. Pre-exposed gilts received antibiotic treatment to clear the *Cs* infection; then, pigs were used to investigate the effect of vaccination on (A) chlamydial burden and (B) the induction of humoral and cellular immune responses. 

### 3.1. Chlamydia Load in Vaginal Swabs 

#### 3.1.1. Pre-exposure and the efficacy of the antibiotic treatment

Prior to and after antibiotic treatment, rectal and genital *Cs* loads were determined from rectal and vaginal swabs to confirm that the antibiotic treatment was successful in clearing ongoing *Cs* infections. Prior to antibiotic treatment, all animals (24/24) were *Cs* positive in the rectum with a median *Cs* load of 1647 *Cs* particles per swab ( = Cs/swab). In the genital tract, one third of the animals (8/24) were *Cs* positive; within these, the median *Cs* load was 49 *Cs*/swab. In contrast, post antibiotic treatment, pigs had either cleared (12/24) or vastly reduced (12/24) the gastrointestinal tract *Cs* infection (Median of *Cs*-infected pigs: 5 *Cs* /swab). Most importantly, all pigs had cleared the genital tract *Cs* infection (Appendix A).

#### 3.1.2. Effect of Vaccination on the Genital *Cs* Load 

After clearance of prior genital *Cs* infections, pigs were vaccinated at 0 and 14 dpv. At 40 dpv (equaling 0 dpc), pigs were challenged post-cervically with 10^8^
*Cs* particles. Vaginal *Cs* shedding assessed by qPCR was used to determine vaccine efficacy. The effect of vaccination on the genital *Cs* load is shown in Figure 3. Prior to challenge, pigs from all groups were negative for *Cs*: MOCK (gray), *Cs* challenged (*Cs*-chall, blue), *Cs* challenged + vaccinated (*Cs*-chall + vacc, orange), and *Cs* challenged + Tri Adjuvant vaccinated (*Cs*-chall + TriAdj vacc, red). In addition, all MOCK challenged animals stayed negative throughout the study. All but one of the *Cs*-challenged pigs developed an active genital infection starting at 1 dpc, peaking at 2 dpc and declining by 4 dpc. While two–three pigs from the *Cs*-chall group stayed *Cs* positive until 7 dpi, all pigs in the *Cs*-chall + vacc group had cleared the *Cs* infection at this time point, and all pigs in the *Cs*-chall + TriAdj vacc group had cleared the *Cs* infection by 6 dpi (data not shown). Non-vaccinated animals (blue) showed the strongest *Cs* propagation with a peak median abundance of 8685 *Cs*/swab. Both vaccinated groups had a significantly decreased genital chlamydial burden compared to *Cs* challenged: the non-adjuvanted group (orange) peaked at 1661 *Cs*/swab; the TriAdj-adjuvanted group peaked at 2228 *Cs*/swab. This vaccine-induced reduction shows the efficacy of both vaccines in reducing the genital *Cs* burden.

### 3.2. The Humoral Immune Response to Chlamydia suis Vaccination and Challenge

After showing that both vaccines reduced the post-challenge *Cs* burden, we investigated the mechanisms involved in the anti-chlamydia immune response. The *Cs* humoral immune response was evaluated in serum in two ways—by determining the anti-*Cs* IgG levels using *Cs*-specific multi peptide ELISAs (Figure 4A), and the effect of neutralizing antibodies on suppression of *Cs* infection in HeLa cells (Figure 4B). Except for a trend to higher serum IgG levels at 40 dpv in the vaccinated animals, no statistical differences were observed for the humoral immune response upon *Cs* vaccination and/or challenge. In addition, day 0 sera already showed high IgG titers (O.D. ~1.0) and neutralizing antibody levels that suppressed on average ~90% of *Ct* (Figure 5). These data document pre-existing humoral anti-*Cs* immunity in commercial high-health farm raised pigs.

### 3.3. The CD4 T-Cell Response to Chlamydia suis Vaccination and Challenge 

In addition to the humoral immune response, we analyzed the CD4^+^ T-cell response, which is the most important adaptive immune response against chlamydia. PBMCs were re-stimulated in vitro with *Cs* lysate to determine the anti-*Cs* CD4 T-cell response via multicolor flow cytometry. The proliferative response and IFN-γ production of CD4 T cells as well as the differentiation of these responding cells from naïve (T_naïve_) into lymph node trafficking central memory (T_CM_) and tissue-trafficking effector memory (T_EM_) cells is shown in Figure 5. At the time points with the highest CD4 T-cell proliferation (40 dpv) and IFN-γ production (37 dpv), we performed two novel analyses in addition to the standard percentage analysis of the cellular response (Proliferation, Figure 5A; IFN-γ production, Figure 5E). As described in the Materials and Methods (Section 2.8), we calculated a “differentiation value” (Figure 5C,G) and a “response value” (Figure 5D,H). The “differentiation value” assesses the differentiation status of the responding cells. The “response value” represents a combined analysis of the percentage of responding cells (proliferating or IFN-γ^+^ CD4 T cells) and the differentiation value of the responding cells. 

The proliferative response of CD4^+^ T cells to *Cs* is shown in Figure 5A–D. Even before vaccination at 0 dpv, CD4 T cells from these pre-exposed pigs showed a strong proliferative response to *Cs*. This response dropped until 33 dpv. While T cells from pigs from both non-vaccinated groups (MOCK and *Cs*-chall) exhibited low responses at 37 dpv (=7 dpc), both vaccinated groups showed an increase in their T-cell proliferative response at this time point. At 40 dpv, or 10 dpc, T cells from all challenged pigs proliferated more strongly to *Cs* than T cells from MOCK animals (Figure 5A). These data show that *Cs* infection leads to a strong proliferative CD4 T-cell response. However, there was no significant difference in the proliferative response between the *Cs*-chall and the vaccinated groups.

The differentiation of the CD4 T cells that proliferate upon *Cs*-restimulation at 40 dpv is shown in Figure 5B. While in MOCK pigs the majority (~60%) of proliferating CD4 T cells were naïve, only 38% of these cells belonged to the T_CM_ fraction. In contrast, in all three *Cs*-challenged groups there was an even distribution between naïve and T_CM_ proliferating CD4 T cells (50–53%). Interestingly, *Cs*-specific CD4^+^ T_EM_ cells did not show a significant proliferative response in any of the groups. 

Differentiation analyses using the differentiation value at 40 dpv revealed that, compared with MOCK animals, proliferating CD4 T cells from *Cs*-chall were more differentiated (Figure 5C, *p* = 0.04). While the response value did not reach a significant difference between the MOCK and either of the challenged groups, all challenged groups had by number a higher response value (Figure 5D). In summary, *Cs* infection, either by pre-exposure to *Cs* or *Cs* challenge during the trial, induced a strong proliferative response in CD4 T cells, and by trend, it primed for a higher frequency of differentiated proliferating CD4 T cells.

To provide more details on the systemic anti-chlamydia response of CD4 T-cells induced by *Cs* infection, we performed an in-depth analysis of their transcriptional profile using RNAseq. PBMC were restimulated overnight with *Cs* and their transcriptional profile was compared between MOCK and *Cs*-chall pigs at 37 dpv, based on the peak IFN-γ response at this time point (Figure 5). The data are summarized in Appendix A: Compared to MOCK controls, 89 transcripts were significantly (*p* < 0.05) upregulated in CD4 T cells isolated from the blood of pigs trans-cervically challenged with *Cs*. These genes are primarily involved in T-cell growth, proliferation, adhesion, migration, inflammation, and immunity: We observed the upregulation of key genes involved in T-cell receptor (TCR) signaling and activation, including *PKD2* [32,33], *VDR* [34], *PFDN1* [35], and *TRAF3* [36], along with genes important for T cell survival, growth, and proliferation mediated via the AKT signaling pathway such as *IGF1* [37], *TNFSF11* [38,39], and *PINK1* [40]. We also observed upregulation of *RORA* and *DAPK1*, which are critical for T-cell differentiation and effector function [41,42,43,44]. These transcriptional profiling data can provide valuable insight into the details of the systemic anti-chlamydia CD4 T-cell response, and they will be used as the basis for future more in-depth analyses on this crucial immune response in the highly relevant pig model.

In this study, we focused our analysis on the CD4 T-cell response that has been shown to be most crucial for protection against chlamydia—their IFN-γ response and their ability to migrate to the genital tract tissue. Studying the IFN-γ response of CD4 T cells revealed a contrast to the above-described proliferation data: despite pre-exposure to *Cs*, IFN-γ production by CD4 T cells was low in all groups at the start of the trial (0 dpv). Within all groups, the median frequency of IFN-γ^+^ CD4 T cells stayed below 0.2% at 0 dpv (Figure 5E); this response stayed low through 33 dpv. However, at 37 dpv (7 dpc), compared to *Cs*-chall animals, the IFN-γ response from CD4 T cells of both vaccinated groups increased either by number (*Cs*-chall + vacc) or significantly (*Cs*-chall + TriAdj vacc, Figure 5E, *p* < 0.0001).

Figure 5F–H shows the differentiation of these IFN-γ-producing CD4 T cells at their peak response (37 dpv). Compared with *Cs*-chall animals, IFN-γ^+^ CD4 T cells from the TriAdj vaccinated animals had more memory cells with the majority of them being tissue-trafficking T_EM_ cells (Figure 5F). As a result, these cells also had a higher differentiation value in TriAdj vaccinated pigs compared to *Cs*-chall pigs (Figure 5G, *p* = 0.03). Compared to the *Cs*-chall group, the combinatorial response value was also significantly increased in the *Cs*-chall + TriAdj vacc group (Figure 5H, *p* = 0.02). This comparison shows the effect of the adjuvanted vaccine on both IFN-γ production and differentiation of the responding CD4 T cells. In summary, intranasal *Cs* vaccination, especially if adjuvanted with the TriAdj adjuvant, primed for a stronger IFN-γ response post challenge; moreover, it induced the differentiation of CD4 T cells into tissue-trafficking T_EM_ cells.

## 4. Discussion

Due to the high prevalence of *Ct* in humans, vaccination of pre-exposed patients would benefit a timely establishment of herd immunity. Therefore, the goal of this study was to provide an animal model to test future *Ct* vaccine candidates in outbred pigs naturally pre-exposed to *Cs*—a close relative of *Ct*. This model has strong potential to predict *Ct* vaccine safety and efficacy for both critical Phase III clinical trials in a high-risk population and pre-exposed *Ct* patients. 

To establish this model, we performed a proof-of-principle experiment studying *Cs* vaccine efficacy and immunogenicity in *Cs* pre-exposed pigs. At arrival, all pigs were pre-exposed to *Cs* as demonstrated by anti-*Cs* serum IgG levels. In addition, 100% of pigs had an ongoing infection in the gastrointestinal tract, and 33% were *Cs*-positive in the genital tract. Antibiotic treatment with doxycycline and tylosin cleared the *Cs* infection in 50% of the GI tracts and vastly reduced the *Cs* burden in the remaining 50%. This treatment also cleared 100% of genital tract *Cs* infections (Appendix A). This demonstrates the efficacy of this combinatorial treatment plan for *Cs* in pigs. Additionally, in turn, this substantial reduction of *Cs* from these pigs explains a peculiarity of the observed immune response—the drop in antibody levels and CD4 proliferation from 0 dpv to 24 dpv even in vaccinated pigs. While the antibiotic treatment nearly eliminated *Cs* from these pigs, it is unlikely that the humoral and T-cell response vanishes within 5 days. This is reflected by the substantial serum antibody levels (Appendix A) and the high frequency of proliferating CD4 T cells (Figure 5A) at 0 dpv (5 days post antibiotic treatment). However, this antibody and proliferative response dropped substantially by 24 dpv (29 days post antibiotic treatment). This drop shows that the humoral and T-cell response to natural *Cs* infection strongly declined within one month of antibiotic treatment. This drop was present in all groups, thus, neither of the vaccines could overcome the effect of *Cs* clearance on the systemic antibody levels and CD4 T-cell proliferation. In summary, these data show that in contrast to our vaccine candidates, natural *Cs* infection induces a strong systemic humoral immune response and CD4 T-cell proliferation which declines within one month after antibiotic treatment.

While natural infection led to high serum antibody levels and a strong proliferative CD4 response, it did not induce a notable IFN-γ response (Figure 5). Since genital infections with both *Cs* and *Ct* induce an IFN-γ response [10,11,20], the observed lack of IFN-γ production despite a clear proliferative response can potentially be explained by the fact that all pre-exposed pigs were mainly infected in the GI tract; however, chlamydia infections of the GI tract have been shown to be rather homeostatic [45]. Since IFN-γ production by CD4 T cells is a central immune mechanism in the clearance of both *Cs* and *Ct*, this result indicates that GI *Cs* and *Ct* infections are not likely to be cleared by the natural immune response and require antibiotic treatment. 

The final question that needs to be addressed is: which immune mechanism did the vaccines induce to fasten the time to recovery and significantly lower the genital *Cs* load on days 2 and 3 post challenge (Figure 3)? To answer that question, we studied the humoral and CD4 T-cell response pre-vaccination (0 dpv), at 10 days post boost vaccination (24 dpv), and at 3, 7, and 10 days post challenge (33, 37, and 40 dpv, respectively). While both vaccines reduced the genital *Cs* burden and shortened the time to *Cs* clearance, neither of the vaccines induced a significant systemic humoral immune response (Figure 4) or a significant CD4 T-cell response (Figure 5) at 24 dpv. This lack of vaccine-induced systemic pre-challenge immunity can be explained in three ways. First, the high blood antibody levels and CD4 T-cell proliferation at 0 dpv might mask a potential induction of these immune parameters. Second, while 10 days post infection was an adequate time point in a previous *Cs*/*Ct* challenge study [11], 24 dpv (thus, 10 days post boost vaccination) might have been too late for the detection of vaccine-induced CD4 IFN-γ production. Post-challenge, this response peaked at 7 dpc and already declined by 10 dpc; this indicates that 7 but not 10 days post boost vaccination might have revealed a vaccine-induced CD4 T-cell response. Third, the chosen intranasal route of vaccination has been shown to induce a strong mucosal anti-chlamydia response [7,46]; while this response is relevant for protection, the induction/boost of a systemic immune response might be limited. An intramuscular/intranasal prime-boost regimen might have been superior in the induction of a systemic humoral and cellular response as shown by a recent clinical phase I *Ct* vaccine trial [47].

Post challenge, vaccination did not induce a significantly enhanced systemic humoral response either. This is in line with our results from a previous *Cs* and *Ct* infection study [11]. Nevertheless, this result does not rule out a contribution of the local humoral immune response in the protection against genital *Ct* infection as shown by Erneholm et al. [20].

While the humoral immune response was not altered significantly by *Cs* vaccination, both vaccinations primed for a stronger and arguably more efficient post-challenge CD4 T-cell response. Intranasal administration of UV-inactivated *Cs* particles with and without TriAdj adjuvant improved the CD4 T-cell response in three ways: first, although only by number, compared to the MOCK group, both vaccinated groups primed for a faster proliferative response of CD4 T cells (Figure 5A, 37 dpv). Second, compared with the *Cs*-chall group, the TriAdj-adjuvanted vaccine induced a significantly stronger post-challenge IFN-γ response (Figure 5E, 37 dpv). Third, both vaccines, but mainly the TriAdj vaccine, induced the differentiation of IFN-γ-producing CD4 T cells into memory cells, especially into tissue-trafficking T_EM_ cells (Figure 5F–H). In pigs, IFN-γ-producing CD4 T_EM_ cells have been shown to have a crucial role in protection of genital *Ct* infections. First, we showed that pigs develop a strong IFN-γ CD4^+^ T-cell response upon *Ct* or *Cs* infection [11]; second, the influx of CD4 T cells into the genital tract tissue has previously been shown to represent a good correlate of protection: Erneholm et al. showed in minipigs that cervical infiltration of CD4 T cells upon immunization with inactivated *Ct* + CAF01 adjuvant was associated with protection against genital *Ct* infection [20]. Furthermore, the importance of these IFN-γ-producing CD4^+^ T cells is demonstrated by their association with reduced risk of *Ct* re-infection in humans [48]. These cells have also been shown to correlate with protection [49]; moreover, they are essential for pathogen clearance [7,50,51]. During chlamydia infection in mice, naïve T cells differentiate into effector T cells in uterus-trafficking lymph nodes and these effector cells are then recruited to the mucosa of the genital tract to establish tissue-resident memory cells (T_RM_) [21,52,53]. Furthermore, in 2015, Stary et al. showed that the generation of “two waves of protective memory T cells,” tissue-resident CD4 T_RM_ and circulating T_CM_ and T_EM_ cells, is required for optimal clearance of genital *Ct* infections [7]. Based on the combination of i) the protective effect of our vaccines on the genital chlamydial burden and ii) the induction of CD4 T-cell differentiation into tissue-trafficking IFN-γ-producing CD4 T_EM_ cells, our data support two main conclusions of these studies: first, IFN-γ-producing tissue-trafficking CD4 T_EM_ cells, which can further differentiate into T_RM_ cells, could serve as a promising blood biomarker for protection against genital *Ct* infections; second, mucosal delivery of adjuvanted UV-inactivated chlamydia particles is a promising strategy for *Ct* vaccine development.

## 5. Conclusions

This proof-of-principle study demonstrates that intranasal vaccination with UV-inactivated *Cs* particles +/- TriAdj is immunogenic; moreover, it lowers genital *Cs* burden even in pre-exposed outbred pigs. This vaccination primes for a stronger IFN-γ response of CD4 T cells; additionally, it drives CD4 T-cell differentiation into both lymph node trafficking central memory T cells and tissue trafficking effector memory T cells. Thus, this vaccination induces both waves of CD4 T-cell memory—the immune mechanisms previously reported to be required for protection against *Ct* [7,20]. Thereby, this study provides the first insight to our knowledge into the performance of a chlamydia vaccine candidate in pre-exposed outbred animals. This insight supports that mucosal *Ct* vaccination can boost immunity in pre-exposed hosts, which is highly relevant during phase III clinical vaccine trials. This study further supports the relevance of the pig model for translational research on *Ct*. Future studies will target the evaluation of different adjuvants and prime/boost regimens for *Ct* vaccines as well as a direct comparison between vaccine efficacy and immunogenicity in naïve and *Cs*-pre-exposed outbred (mini-) pigs.

## Figures and Tables

**Figure 1 vaccines-08-00353-f001:**
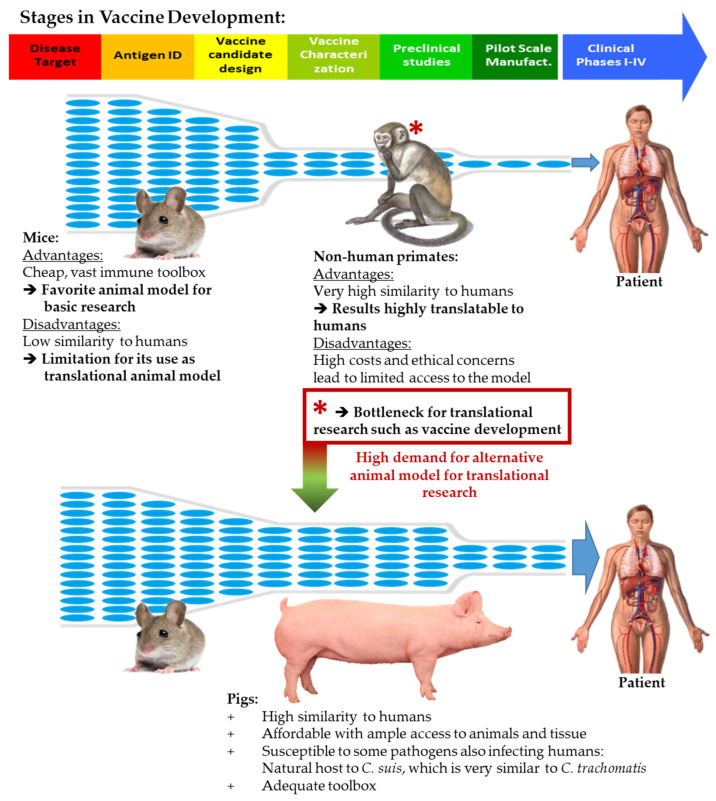
Animal models for vaccine development. The various animal models have advantages and disadvantages during the process of vaccine development. While the associated costs and vast immune toolbox of mice make them an excellent model for basic research, non-human primates are an excellent choice for translational research. Due to high costs and ethical concerns, access to non-human primates is limited, creating a bottleneck for vaccine development. The pig has several advantages as a biomedical translational animal model and can open the bottleneck to advance promising vaccine candidates into clinical trials. The stages in vaccine development were adapted from https://www.niaid.nih.gov/research/vaccine-development-pipeline.

**Figure 2 vaccines-08-00353-f002:**
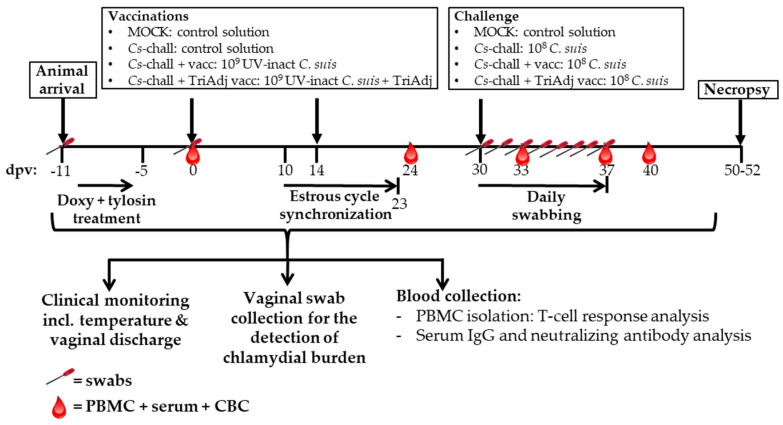
Setup of the in vivo *C. suis* vaccination experiment. Twenty-four sexually mature female *C. suis* pre-exposed pigs were randomly distributed into four groups with six pigs each. Pigs were treated with doxycycline and tylosin from 11 to 5 days post vaccination (dpv). At 0 and 14 dpv, pigs were vaccinated intranasally with: control solution (MOCK and *Cs*-chall), 10^9^ UV-inactivated *C. suis* inclusion-forming units (IFU) (*Cs*-chall + vacc), or 10^9^ UV-inactivated *C. suis* IFU + Tri Adjuvant (*Cs*-chall + TriAdj vacc). From 10–23 dpv, the estrus cycle of pigs was synchronized with synthetic progesterone. At 30 dpv, pigs were challenged trans-cervically with control solution (MOCK) or 10^8^
*C. suis* IFU (*Cs*-chall, *Cs*-chall + vacc and *Cs*-chall + TriAdj vacc). Pigs were clinically monitored every day throughout the study. Blood and vaginal swab collection were performed as indicated on the figure by the blood drop and the swab symbol. Necropsy was performed on 50–52 dpv. *C. suis* were detected in vaginal swabs via qPCR. Both IgG and neutralizing antibody levels were determined in serum. Peripheral blood mononuclear cell (PBMC) were used for analysis of T-cell mediated immune responses.

**Figure 3 vaccines-08-00353-f003:**
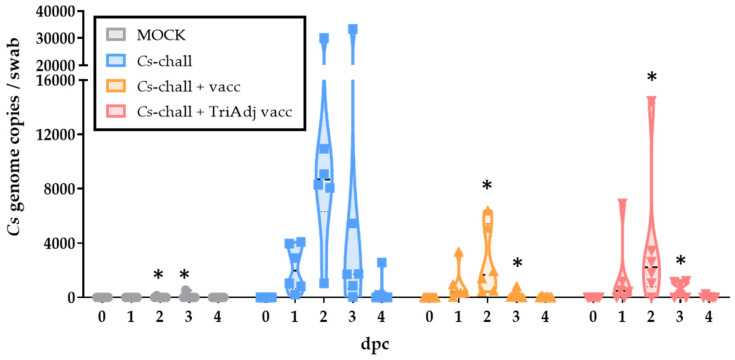
*C. suis* load is reduced in vaccinated pigs post challenge. *C. suis* load was analyzed via qPCR in vaginal swabs from MOCK (gray), *C. suis* challenged (*Cs*-chall, blue), *C. suis* challenged + vaccinated (*Cs*-chall + vacc, orange), and *C. suis* challenged + Tri Adjuvant vaccinated (*Cs*-chall + TriAdj vacc, red) pigs prior to challenge (0 days post challenge, dpc) and after challenge (1–4 dpc). Values for individual pigs, medians, and 25/75 percentiles are shown. Data were analyzed using a repeated-measures two-way ANOVA in comparison to *Cs*-chall animals and corrected for multiple comparisons with Dunnett’s multiple comparisons test. * *p* < 0.05.

**Figure 4 vaccines-08-00353-f004:**
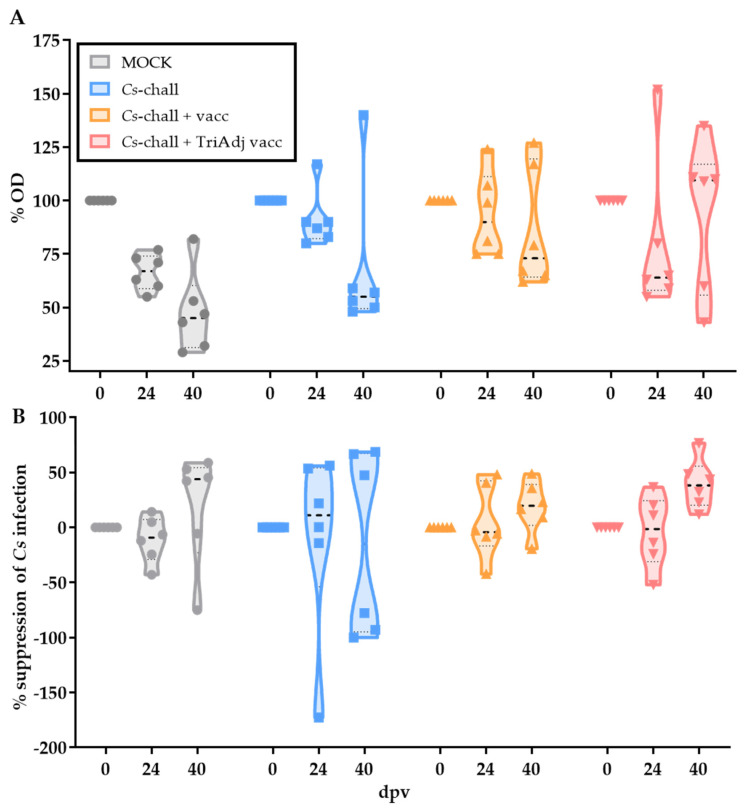
The systemic anti-*Cs* humoral immune response is equivalent in pre-exposed, challenged, and vaccinated/challenged pigs. The humoral immune response for each individual pig was analyzed in MOCK (gray), *C. suis* challenged (*Cs*-chall, blue), *C. suis* challenged + vaccinated (*Cs*-chall + vacc, orange), and *C. suis* challenged + Tri Adjuvant vaccinated (*Cs*-chall + TriAdj vacc, red) pigs. (**A**) Anti-*C. suis* serum IgG levels at stated days post first vaccination (dpv, x-axis) were analyzed via peptide ELISA. While no significant differences were obtained, serum anti-*Cs* IgG levels were elevated by number at 40 days post vaccination (10 days post challenge) in the TriAdj vaccinated pigs. (**B**) Neutralizing antibodies in serum were analyzed by infecting HeLa cells with *Cs* in the presence of serum (1:10 dilution). The % suppression of infection (y axis) was calculated via flow cytometry by dividing the infection rate at the day post first vaccination (dpv, x-axis) by the infection rate before challenge within the same animal. No between-group differences were obtained. Values for individual pigs, medians, and 25/75 percentiles are shown.

**Figure 5 vaccines-08-00353-f005:**
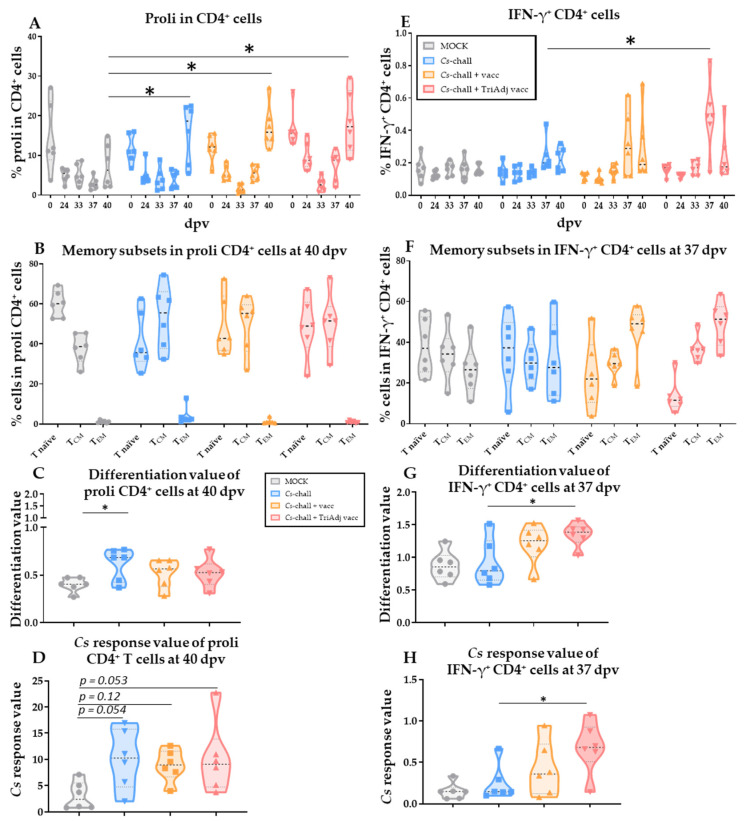
The CD4 T-cell response to *C. suis*. While *C. suis* infection induces CD4 T-cell proliferation, intranasal vaccination drives CD4^+^ T-cell maturation and primes for a stronger IFN-γ response and post-challenge. PBMC from the stated days post first vaccination (dpv) were isolated from MOCK (gray), *C. suis* challenged (*Cs*-chall, blue), *C. suis* challenged + vaccinated (*Cs*-chall + vacc, orange), and *C. suis* challenged + Tri Adjuvant vaccinated (*Cs*-chall + TriAdj vacc, red) pigs. For *C. suis*-specific proliferative response analysis (**A**,**D**), PBMC were stained with CellTrace^TM^ Violet and cultured with *C. suis* lysate for four days. For *C. suis*-specific IFN-γ production analysis (**E** to **H**), PBMC were cultured with *C. suis* lysate for 18 h in the presence of a Golgi inhibitor for the last 4 hours. Cells were then harvested and stained as indicated in Table 1. Proliferating CD4^+^ (proli CD4^+^) or IFN-γ^+^ CD4^+^ cells were further differentiated as T_naïve_, T central memory (T_CM_) and T effector memory (T_EM_) at specific dpv (**B**,**F**). Differentiation and response values were calculated for proliferating CD4^+^ cells (**C**,**D**, respectively) and for IFN-γ^+^ CD4^+^ cells (**G**,**H**, respectively). Values for individual pigs, medians, and 25/75 percentiles are shown. Data were analyzed using repeated measures two-way ANOVA in comparison to MOCK (**A**,**B**) and *Cs*-inf animals (**E**,**F**) with Dunnett’s multiple comparisons test, and with one-way ANOVA in comparison to MOCK (**C**,**D**) and *Cs*-chall animals (**G**,**H**). * *p* < 0.05.

**Table 1 vaccines-08-00353-t001:** Flow cytometry antibody panels.

Antigen	Clone	Isotype	Fluorochrome	Labeling Strategy	Primary Ab Source	2nd Ab Source
	Peripheral blood mononuclear cell (PBMC) proliferation staining panel					
CD3	PPT3	IgG1	FITC	Directly conjugated	Southern Biotech	-
CD4	74-12-4	IgG2b	Brilliant Violet 480	Secondary antibody	BEI Resources	Jackson Immunoresearch
CD8α	76-2-11	IgG2a	Brilliant Violet 605	Biotin-streptavidin	Southern Biotech	Biolegend
CCR7	3D12	rIgG2a	Brilliant Blue 700	Directly conjugated	BD Biosciences	-
Live/Dead	-	-	Near Infra-red	-	Invitrogen	-
Proliferation	-	-	CellTrace^TM^ Violet	-	Invitrogen	-
	*PBMC Intracellular cytokine staining panel*					
CD3	PPT3	IgG1	FITC	Directly conjugated	Southern Biotech	-
CD4	74-12-4	IgG2b	Brilliant Violet 421	Secondary antibody	BEI Resources	Jackson Immunoresearch
CD8α	76-2-11	IgG2a	PE-Cy5.5	Biotin-streptavidin	Southern Biotech	Southern Biotech
CCR7	3D12	rIgG2a	Brilliant Violet 480	Directly conjugated	BD Biosciences	-
IFN-γ	P2G10	IgG1	PE	Directly conjugated	BD Biosciences	-
Live/Dead	-	-	Near-Infrared	-	Invitrogen	-
	*PBMC MACS reanalysis staining panel*					
CD4	74-12-4	IgG2b	PE	Secondary antibody	BEI Resources	Southern Biotech
CD172a	74-22-15	IgG1	Alexa Flour 647	Secondary antibody	BEI Resources	Southern Biotech
Live/Dead	-	-	Near Infra-red	-	Invitrogen	-

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
