# Peer review of "Mucosal Vaccination with UV-Inactivated Chlamydia suis in Pre-Exposed Outbred Pigs Decreases Pathogen Load and Induces CD4 T-Cell Maturation into IFN-γ+ Effector Memory Cells"

_vaccines, 2020, doi:10.3390/vaccines8030353_

Round 1

Reviewer 1 Report

In this study, the authors analyzed the immunogenicity and efficacy of Chlamydia suis vaccine candidates in pigs which have been previously pre-exposed to Chlamydia suis. Female pigs infected with Chlamydia suis are first treated with antibiotics to eradicate bacterial infection. Then, animals (6 pigs per group) either received 2 injections of UV-inactivated Chlamydia suis particles in the presence or absence of an adjuvant or a MOCK injection. 30 days post-vaccination, pigs were challenged trans-cervically with Chlamydia suis. Authors show that both vaccinated groups had a decreased genital chlamydial burden compared to unvaccinated animals. Authors then analyzed immune response parameters. They show that vaccination of Chlamydia suis pre-exposed animals has no impact on the humoral response. Indeed, unvaccinated animals have already a pre-existing humoral anti-Chlamydia immunity characterized by a high level of neutralizing IgG antibodies in the blood. Vaccination does not enhance this humoral response. Authors then analyzed the CD4+ T-cell response and conclude that the 2 vaccine candidates induce IFN-γ secreting CD4+T cells and the differentiation of these cells into tissue-trafficking TEM cells.

In this study, experiments are well-conducted and a lot of work has been done. It clearly demonstrates that pigs are a suitable model for testing vaccines in the context of Chlamydia infections. However, I am not convinced by the conclusions of the authors whom should bring more information and data to reinforce their study.

First of all, I do not totally understand the relevance of the study. Authors analyze the effect of vaccines in the context of pre-exposed pigs that have already developed immunity against Chlamydia.They argue that wide vaccine coverage is necessary to acquire a herd immunity against Chlamydia and that due to the high prevalence of  Chlamydia, it would not be sufficient to vaccinate only naïve patients. This argument is relevant if pre-exposed individuals did not develop a good immunity and if the vaccine improves this immunity. According to me, the results of the study do not show a real effect of vaccines in their pig’s models because of a pre-existing immunity that seems to be sufficient to control a new Chlamydia infection.

Indeed, in figure 3, the authors measured the bacterial load several days after the bacterial challenge. They observe a peak of the bacterial load at day 2 post-challenge, as well as in unvaccinated and in vaccinated animals but the load is 4 fold less important for vaccinated animals. Importantly, 4 days post-challenge, bacterial load decreases and they is no difference anymore between unvaccinated and vaccinated animals. This result shows that pre-exposed and unvaccinated animals naturally control Chlamydia suis infection almost as quickly as vaccinated animals (since as soon as day 3  the number of Cs genome copies/swab dramatically decreases) so the utility of vaccines is questionable. Perhaps the control of infection can depend on the initial bacterial load used for the challenge. How do the authors choose this load? Is the bacterial infection completely cleared after day 4 post-challenge? Is there a real benefit of vaccination? Is the intranasal vaccination the best route of immunization against Chlamydia?

In the same way, authors show in figure 4 that there is no difference in the humoral response between unvaccinated and vaccinated animals because of an existing pre-immunity in animals. So, what is the interest to vaccinate these animals?

Concerning the CD4 T cell responses, authors show a difference in the percentage of IFN γ+ secreting CD4+ T cells between vaccinated animals and unvaccinated animals but this is true only at 37 days post-vaccination (7 days post-challenge). Moreover, this difference appears after the clearance of the bacterial infection (4 days post-challenge) so how the authors explained the antibacterial effect of their vaccines? One more time, the effect of vaccination in these animals seems to be reduced.

Analyzing the effect of vaccines in pre-exposed animals is interesting but authors should moderate their conclusions because the beneficial vaccinal effect is not clear in this study. To reinforce this work, it would be attractive to analyze the efficacy of their vaccines also in naïve animals never exposed to Chlamydia.

Author Response

Vaccines 835437: Revision 1 – Response to reviewer 1:
Comment 1: In this study, the authors analyzed the immunogenicity and efficacy of Chlamydia suis vaccine candidates in pigs which have been previously pre-exposed to Chlamydia suis. Female pigs infected with Chlamydia suis are first treated with antibiotics to eradicate bacterial infection. Then, animals (6 pigs per group) either received 2 injections of UV-inactivated Chlamydia suis particles in the presence or absence of an adjuvant or a MOCK injection. 30 days post-vaccination, pigs were challenged trans-cervically with Chlamydia suis. Authors show that both vaccinated groups had a decreased genital chlamydial burden compared to unvaccinated animals. Authors then analyzed immune response parameters. They show that vaccination of Chlamydia suis pre-exposed animals has no impact on the humoral response. Indeed, unvaccinated animals have already a pre-existing humoral anti-Chlamydia immunity characterized by a high level of neutralizing IgG antibodies in the blood. Vaccination does not enhance this humoral response. Authors then analyzed the CD4+ T-cell response and conclude that the 2 vaccine candidates induce IFN-γ secreting CD4+T cells and the differentiation of these cells into tissue-trafficking TEM cells.
Response 1: Thank you for the concise and correct summary of our study.
Comment 2: In this study, experiments are well-conducted and a lot of work has been done. It clearly demonstrates that pigs are a suitable model for testing vaccines in the context of Chlamydia infections.
Response 2: Thank you for acknowledging the adequate selection and conduction of the performed experiments.
Comment 3: However, I am not convinced by the conclusions of the authors who should bring more information and data to reinforce their study.
Response 3: This summarizing comment is explained in detail below. We will therefore address these details in the adequate positions.
Comment 4: First of all, I do not totally understand the relevance of the study. Authors analyze the effect of vaccines in the context of pre-exposed pigs that have already developed immunity against Chlamydia. They argue that wide vaccine coverage is necessary to acquire a herd immunity against Chlamydia and that due to the high prevalence of Chlamydia, it would not be sufficient to vaccinate only naïve patients. This argument is relevant if pre-exposed individuals did not develop a good immunity and if the vaccine improves this immunity. According to me, the results of the study do not show a real effect of vaccines in their pig’s models because of a pre-existing immunity that seems to be sufficient to control a new Chlamydia infection. Indeed, in figure 3, the authors measured the bacterial load several days after the bacterial challenge. They observe a peak of the bacterial load at day 2 post-challenge, as well as in unvaccinated and in vaccinated animals but the load is 4-fold less important for vaccinated animals. Importantly, 4 days post-challenge, bacterial load decreases and there is no difference anymore between unvaccinated and vaccinated animals. This result shows that pre-exposed and unvaccinated animals naturally control Chlamydia suis infection almost as quickly as vaccinated animals (since as soon as day 3 the number of Cs genome copies/swab dramatically decreases) so the utility of vaccines is questionable.
Response 4: We thank the reviewer for this critical and constructive comment. This comment led us to re-evaluate our infection data which revealed that vaccination not only significantly decreased the chlamydial burden at 2 and 3 days post challenge (dpc), but it also led to a faster time to recovery: While 2-3 out of 6 animals in the non-vaccinated Cs-chall group stayed positive until 7 dpc, all animals in the vaccinated groups cleared the infection by 7 dpc (Cs-chall + vacc) or 6 dpc (Cs-chall + TriAdj vacc). In addition to the faster clearance in vaccinated animals, this infection kinetic of control animals is very
similar to results obtained in naïve minipigs as shown by Erneholm et al., 2019 in which 8/9 control animals were C. trachomatis positive at 3 dpc and only 2/9 control animals stayed C. trachomatis positive until 7 dpc.
While we still think that focusing Figure 3 on the most relevant time points (0-4 dpc) with significant differences is the optimal presentation of these data, we included a statement on the course of infection past 4 dpi summarizing the prolonged C. suis infection and the shorter time to recovery in the vaccinated animals in Lines 366-368 and 549. Thereby, thanks to the reviewers comment, we now better demonstrate vaccine efficacy and the relevance of the pre-exposed outbred pig model for chlamydia vaccine development – the (to our knowledge) unique large animal model with the ability to mimic phase III clinical trials.
Comment 5: Perhaps the control of infection can depend on the initial bacterial load used for the challenge. How do the authors choose this load? Is the bacterial infection completely cleared after day 4 post-challenge? Is there a real benefit of vaccination? Is the intranasal vaccination the best route of immunization against Chlamydia?
Response 5: The challenge dose was established in previous titration experiments performed by Dr. Kaeser. In addition, the challenge dose represents a standard dose for genital challenge of (mini-) pigs (Vanrompay et al, 2005; Schautteet et al, 2011; Schautteet et al, 2012; Lorenzen et al, 2017). An according statement was included in the manuscript (L. 214).
The clearance of bacterial infection and benefit of vaccination has been addressed above.
We chose this intranasal/intranasal prime boost regimen due to the literature that shows not only a crucial role for tissue-trafficking CD4 T cells in the protection against chlamydia but also that intranasal vaccination successfully induces these protective tissue-trafficking CD4 T cells. Nevertheless, we cannot rule out that another prime-boost regimen could improve vaccine efficacy. Indeed, such a direct comparison of different prime-boost regimens (intramuscular/intramuscular, intramuscular/intranasal, intranasal/intranasal) is planned for future studies upon successful funding acquisition. We included a statement on a potential superior performance of an IM/IN vaccine regimen in lines 563-565 and a reference to the above-mentioned future studies in the manuscript (L. 610-613).
Comment 6: In the same way, authors show in figure 4 that there is no difference in the humoral response between unvaccinated and vaccinated animals because of an existing pre-immunity in animals. So, what is the interest to vaccinate these animals?
Response 6: As mentioned above, the infection kinetic of genital chlamydia infections in our outbred pre-exposed pig model is comparable to other established pig models for chlamydial vaccine development. In addition, the unique ability to predict a vaccine outcome in phase III clinical trials provides a strong relevance to this pre-exposed outbred pig model. Indeed, while IgG and IgA in the genital tract correlates with protection against Ct (Erneholm et al., 2019), the role of the systemic humoral immune response is less clear. Therefore, based on these studies and the data presented in this manuscript, the detection of neutralizing antibodies in the blood does not preclude an active chlamydial infection in the genital tract. In addition, as mentioned in Lines 568-570, the lack of induction of a systemic humoral response by the vaccines does not rule out a contribution of the local immune response in the protection against Ct as shown by Erneholm et al., 2019.
Comment 7: Concerning the CD4 T cell responses, authors show a difference in the percentage of IFN γ+ secreting CD4+ T cells between vaccinated animals and unvaccinated animals but this is true only at 37 days post-vaccination (7 days post-challenge). Moreover, this difference appears after the clearance of the bacterial infection (4 days post-challenge) so how the authors explained the antibacterial effect of their vaccines? One more time, the effect of vaccination in these animals seems to be reduced.
Response 7: The reviewer addresses here a critical point: Why did we detect only a vaccine-induced CD4 T-cell response post-challenge, and even post-clearance of infection? We included an extended discussion of this critical point in lines 552-
567). In summary, we have three hypotheses: i) The pre-existing systemic antibody levels and CD4 T-cell proliferation might mask the detection of an increase in these immune parameters by vaccination. ii) the time point for the pre-challenge analysis (24 dpv or 10 days post boost) might have been too late to detect an increased CD4 IFN-γ response. And iii) the intranasal route of administration is mostly inducing a mucosal immune response which explains the limited differences in the systemic response with mostly the induction of tissue-trafficking IFN-γ producing CD4 T cells.
Comment 8: Analyzing the effect of vaccines in pre-exposed animals is interesting but authors should moderate their conclusions because the beneficial vaccinal effect is not clear in this study. To reinforce this work, it would be attractive to analyze the efficacy of their vaccines also in naïve animals never exposed to Chlamydia.
Response 8: We thank the reviewer for this recommendation. We adjusted our conclusion statement (L. 607-609) and as mentioned previously, we included that the direct comparison between vaccines in naïve and pre-exposed animals will be part of future studies upon successful funding acquisition.

Reviewer 2 Report

The study provides an interesting model for evaluate chlamydia vaccine candidate in pre-exposed outbred animals. The model could help the development of new vaccine in human.

The method part could be enhanced to be more easy to read. Figure 2 should be in high image quality.

The authors have to include in the discussion a comparison with human clinical trial recently published.

The use of nasal immunisation only could limit the vaccine efficacy and hence the difference in immune response, could the IM immunisation present a solution for next experiment?

Author Response

Vaccines 835437: Revision 1 – Response to reviewer 2:
Comment 1: The method part could be enhanced to be easily read.
Response 1: We revised the method part using the “Track changes” function to highlight the changes (L. 182/3, 212-214, 220-223, 248-254, 276/7, 307-311).
Comment 2: Figure 2 should be in high image quality.
Response 2: Copy-pasting of the figure from powerpoint into word might have reduced the quality of this figure. In addition to the word document, we provided a zip file with all original powerpoint documents for high-resolution publication. If these original files are still not in sufficient quality, please let us know and we will provide any file format you need.
Comment 3: The authors have to include in the discussion a comparison with human clinical trial recently published.
Response 3: Thank you for your suggestion. We included the reference of the first genital chlamydia vaccine to be tested in humans in lines 563-565 and the reference list.
Comment 4: The use of nasal immunization only could limit the vaccine efficacy and hence the difference in immune response, could the IM immunization present a solution for next experiments?
Response 4: We thank the reviewer for this recommendation. We chose this intranasal/intranasal prime boost regimen due to the literature that shows not only a crucial role for tissue-trafficking CD4 T cells in the protection against chlamydia but also that intranasal vaccination successfully induces these protective tissue-trafficking CD4 T cells. Nevertheless, we cannot rule out that another prime-boost regimen like an IM/IN combination could improve vaccine efficacy. Indeed, such a direct comparison of different prime-boost regimens (intramuscular/intramuscular, intramuscular/intranasal, intranasal/intranasal) is planned for future studies upon successful funding acquisition. We included a statement on a potential superior performance of the vaccine in lines 563-565 and a reference to these future studies in the manuscript (L. 610-613).

Round 2

Reviewer 1 Report

Dear authors, Dear Editor,

As I said in my first report, this study nicely illustrates that pigs are a suitable model for testing vaccines in the context of Chlamydia infections. However, in the first version of the manuscript, the beneficial effect of vaccines was questionable due to the pre-existing immunity in this model of pre-exposed animals. 

Thanks to authors reply and the modifications made in the revised manuscript, it is now much clearer that vaccines accelerate the clearance of bacterial infections and can boost immunity. Therefore the quality of the manuscript and the interest to the reader are greatly improved.